# A Versatile Model for Describing Energy Harvesting Characteristics of Composite-Laminated Piezoelectric Cantilever Patches

**DOI:** 10.3390/s22124457

**Published:** 2022-06-13

**Authors:** Xiaomin Xue, Qing Sun, Qiangli Ma, Jiajia Wang

**Affiliations:** Department of Civil Engineering, Xi’an Jiaotong University, Xi’an 710049, China; sunq@xjtu.edu.cn (Q.S.); 3120322062@stu.xjtu.edu.cn (Q.M.); wjj970226@stu.xjtu.edu.cn (J.W.)

**Keywords:** vibration-based harvesting, composite-laminated piezoelectric patch, cantilever harvester, electromechanical model, optimized parameters

## Abstract

Vibration-based energy harvesters consisting of a laminated piezoelectric cantilever have recently attracted attention for their potential applications. Current studies have mostly focused on the harvesting capacity of piezoelectric harvesters under various conditions, and have given less attention to the electromechanical characteristics that are, in fact, crucial to a deeper understanding of the intrinsic mechanism of piezoelectric harvesting. In addition, the current related models have mostly been suitable for harvesting systems with very specific parameters and have not been applicable if the parameters were vague or unknown. Drawing on the available background information, in this study, we conduct research on a vibration-based cantilever beam of composite-laminated piezoelectric patches through an experimental study of its characteristics as well as a modeling study of energy harvesting. In the experimental study, we set out to investigate the harvesting capacity of the system, as well as the electromechanical (voltage/current-strain and power-strain relationships) characteristics of the cantilever harvester. In addition, we summarize some pivotal rules with regard to several design variables, which provide configuration design suggestions for maximizing energy conversion of this type of harvesting system. In the modeling study, we propose a coupled electromechanical model with a set of updated parameters by using an optimization program. The preceding experimental data are used to verify the superiority of the model for accurately predicting the amount of harvested energy, while effectively imitating the characteristics of a cantilever harvesters. The model also has merit since it is suitable for diversified harvesters with vague or even unknown parameters, which cannot be dealt with by using traditional modeling methods. Overall, the experimental study provides information on a comprehensive way to enhance the harvesting capacity of piezoelectric cantilever transducers, and the modeling study provides a wide scope of applications for cantilever harvesters even if precise information is lacking.

## 1. Introduction

In recent years, vibration-based energy harvesting has attracted attention for its potential use in self-energized, autonomous, and wireless monitoring systems [1,2]. The harvesters are usually classified as piezoelectric [3] electrostatic [4,5], electromagnetic [6,7], magnetostrictive [8,9], electroactive polymer [10], etc. Among the transducers, piezoelectric energy harvesters have received significant attention due to their high energy density, simple fabrication, and their compatibility with MEMS technology [11].

One of most popular harvesters is the piezoelectric cantilever beam harvester using laminated patches that usually consist of one or two piezoelectric layers (unimorph or bimorph) with some other substructures [1,3]. For example, Liu et al. [12] fabricated a MEMS-based piezoelectric laminated cantilever for energy harvesting application integrated with silicon proof mass material, which could realize a quite low resonant frequency. Montanini et al. [13] designed a cantilever-type piezoelectric generator by gluing a PZT patch onto an FRP cantilever beam, and investigated its effect on vibration frequency, vibration amplitude, resistance load, etc. Aramaki et al. [14] fabricated a 2-DOF system using a piezoelectric MEMS vibrational energy harvester with 1-DOF and an Al cantilever, which reached a higher level of output power by 3.4 μW. Usharani et al. [15] presented an energy harvester associated with a piezoelectric-patched cantilever beam that had a step section. It could achieve higher power in a broadband operating frequency. Morimoto et al. [16] fabricated a piezoelectric energy harvester composed of PZT films transferred onto stainless steel cantilevers for improved energy conversion efficiency and toughness.

In addition to the fabrication research on possible applications of such harvesters, researchers have proposed various corresponding mathematical models. Initially, SDOF electromagnetic models [17,18] were presented based on the assumption of a cantilever beam mass-spring-damper system. These modelsprovided simple expressions and were very convenient to use; however, the models were inaccurate because they lacked important aspects of a physical system such as strain distribution and dynamic model shape, as well as their influence on electrical response [19]. As an alternative approach, approximated distributed parameter models [20,21,22] were developed by using Hamilton’s principle and the Rayleigh–Ritz method on the assumption of the Euler–Bernoulli beam theory, which were more accurate approximations as compared with SDOF models. The models mainly considered one single vibration mode and oversimplified the backward coupling in the mechanical domain, and some incorrect modeling attempts easily resulted in inexact solutions [23]. Additionally, Erturk and Inman [19,24] proposed an advanced distributed model to solve its closed-form analytical solution. Its effectiveness was experimentally tested for predicting vibration responses and voltage output. Previous models have targeted energy prediction for harvesters with explicit configurations, and have not been applicable for harvesters with ambiguous or complicated configurations.

Fabrication and modeling studies in the related literature have both basically focused on harvesters’ energy conversion capacities by means of experimental observation and prediction of approximate harvested energy amounts by means of appropriate modeling. For example, most current models have been deemed to be valid through the predictive accuracy of the gross energy amount (such as peak or average electric output). In fact, models should be further tested through electromechanical characteristics that are more pivotal to have a deeper understanding of the piezoelectric harvesting mechanism. Unfortunately, so far, there has not been sufficient model validations. Thus, further detailed model validations are urgently needed. In addition, the current models are mainly applicable for regular or simple piezoelectric-laminated cantilever harvesters (such as unimorph or bimorph) with explicit configurations. However, most harvesting products are encapsulated and complicated (such as MFC and PPA introduced in Section 2), with configurations that are not that simple and explicit. Thus, it is also indispensable to present a model that can versatilely fit for piezoelectric-laminated cantilever harvesters with diversified configurations.

On the basis of the abovementioned background, in this paper, we conduct a series of studies on energy harvesting by using two popular composite-laminated piezoelectric cantilever patches (MFC and PPA) in terms of experimental and modeling studies. In the experimental study, the vibration-based energy harvesting system is set up by using the two types of cantilever patches. Their mechanical and electric output are recorded under different conditions through tuning frequency in exciter, changing resistance in the circuit, and replacing mass weight at the free end of harvesters. On the basis of the experimental results, some crucial parameter settings are summarized for the configuration design of harvesters that maximize energy conversion. Additionally, a full investigation is also executed with regard to the harvesting characteristic relationships including voltage/current-strain and power-strain relationships, which provides a detailed information about the intrinsic electromechanical mechanism. Based on the preceding experimental study, we propose a coupled electromechanical model with set of updated parameters by using an optimization program. The model is verified to be superior through test data, and it is accurate for predicting harvested energy and effective for imitating the characteristic behaviors of the piezoelectric cantilever harvesters. Furthermore, the model can be applied for piezoelectric cantilever harvesters with diversified configurations (even vague or unknown configurations) that cannot be easily dealt with by using traditional modeling methods.

## 2. Energy Harvesting Experiment

For this study, we used two popular commercialized products, i.e., macro fiber composite (MFC) and piezo protection advantage (PPA), as the cantilever beam harvesters and conducted an experiment to investigate their vibration-based harvesting properties. Herein, we briefly introduce the products below.

### 2.1. Harvester Samples

(1)MFC patch

One composite piezoelectric patch, i.e., macro fiber composite (MFC), is a product developed by the NASA Langley Center. It is a layered, planar transducer that employs unidirectional piezoceramic fibers embedded in a thermosetting polymer matrix. The matrix is externally laminated with two polyamide films that are printed interdigited electrodes. The schematic diagram for the construction of the P1 type MFC is shown in Figure 1a, where its capacitance is 1.83 nF, tensile modulus is 30.33 GPa, block force is 162 N, and free strain is 1500.

(2)PPA patch

The other composite piezoelectric patch, i.e., piezo protection advantage (PPA), is one of the piezoelectric unimorph patches produced by the Mide’s company. It is also a laminated transducer but employs a whole layer of piezoceramic in the middle. In addition to the PZT layer, it also has four layers including polyester, copper, steel, and polyamide films. A schematic diagram of its construction is also shown in Figure 1b, where its capacitance is 100 nF, tensile modulus is 30.33 GPa, block force is 0.2 N, and free deflect is 0.8 mm.

### 2.2. Experimental Procedure

As previously mentioned, piezoelectric laminated patches are commonly utilized for cantilever harvesters. Therefore, we used MFC and PPA patches to construct a cantilever harvester with an electric system, which is described below.

As shown in Figure 2, the test system apparatus included the following: a vibration exciter (Bruel & Kjaer Type 4832), a variable resistor (Fumi ZX21 resistance box), data acquisition (QuantumX MX840B module), a signal generator (Bruel & Kjaer Type 2732), and a power amplifier (AWG2005). The vibration exciter was used to vibrate the cantilever MFC/PPA patch; one end of the patch was clamped and connected to the exciter by a harness (see Figure 3). The variable resistor was used to assemble a circuit for carrying an electric current. Data acquisition was adopted to realize multifunctional measuring missions, which simultaneously measured and recorded the voltage (*V*), current (*I*), and strain (S) data from the MFC/PPA. Note that the strain was recorded through a strain gauge pasted at the fixed-center end of the sample. Based on the data collected, the electric power P was calculated by following the electric power relation P=V·I. In Figure 2, MFCs are shown as the testing sample and compensator in the system. In the same way, they were replaced by PPAs when we were about to measure PPA.

### 2.3. Experimental Investigation

By using the setup, series of preliminary tests were carried out to measure the mechanical and electric responses from MFC and PPA under harmonic excitations. Excitation frequency from the exciter was tuned from 1 to 150 Hz with a 5 Hz interval, and the resistance load was changed from 0 to 90 kOhm with a 5 kOhm interval. In the experiment, PPA was also added by a mass weight that could lower the structural dominant frequency. The 4 mass weights chosen were 0 g, 12.5 g, 18.75 g, and 25 g.

#### 2.3.1. Resonant Frequency Determination

It is important to know the natural frequency of a harvester because it dramatically affects the harvesting capability. Thus, a set of sweep frequency tests was executed to determine the natural frequencies of the MFC and PPA structures. Prior to the test, the resistance was maintained at a constant value of 20 kOhm.

First, we increased the excitation frequency slowly from 1 to 150 Hz, and then recorded the strain response. Next, we decomposed the strain data into frequency domain by using Fourier transform, and finally, obtained the single-sided amplitude spectrum sets, as shown in Figure 4.

In Figure 4a, the strain amplitude spectrum for MFC without mass approaches a dramatic peak of 64.85 Hz where resonance is aroused. This indicates that 64.85 Hz is the value of its structural dominant frequency. Similarly, PPA was tested using the four different mass weights. When the PPA cantilever is set without mass, its natural frequency is 123.2 Hz (see Figure 4b). When the mass weight is 12.5 g, the natural frequency changes to a lower value of 31.2 Hz, and the strain amplitude increases drastically to a higher value of 22.59$\;{\mathrm{Hz}}^{-1}$. Next, the same pattern is repeated with mass weights up to 18.75 g and 25 g, the natural frequencies are further decreased to 25.79 Hz and 21.94 Hz, respectively, while the strain amplitude is further magnified to 34.42 ${\mathrm{Hz}}^{-1}$ and 70.06 ${\mathrm{Hz}}^{-1}$.

On the one hand, we found that the maximization of strain and electric power could be aroused, while excitation frequency sweeps are close to the structure’s natural frequency. On the other hand, the natural frequency could be changed, while the tip end had a different weight mass. This indicates there should be an effective way to adopt a suitable structure to accommodate the environment, and thereupon, more energy conversion would be excavated.

#### 2.3.2. Harvesting Capability Comparison

As illustrated in Section 2.3.1, maximization of electric power (harvesting capability) is stimulated when several excitation frequencies approach the dominant frequencies of MFC and PPA cantilever patches, and, in this section, we compare their harvesting capabilities.

Figure 5 shows the results from MFC under excitation with its approximated dominant frequency (60 Hz). Figure 6 shows the corresponding results from PPA under excitation with its approximated dominant frequency (120 Hz). As displayed in Figure 5a, strain response from the fixed-center end of the patch is measured as an amplitude of 1.96 $\times 10^{-3}$. Under this stimulation, the voltage response is then obtained with 1.019 V amplitude, as shown in Figure 5b. Likewise, the current response is obtained as 12.22 μA, as shown in Figure 5c. Based on the voltage and current results, electric power can be calculated based on a simple electric power equation, which is the most important variable associated with vibrational-electric energy conversion. We found that the corresponding peak power was 13.54 μW, as shown in Figure 5d.

Likewise, Figure 6 shows the results from PPA under it dominant frequency excitation (120 Hz). As illustrated in Figure 6b–d, we found the responses’ amplitudes of strain, voltage, current, and power were 0.5442 $\times 10^{-3}$, 8.083 V, 0.8137 mA, and 6.98 mW, respectively.

It should be noted that the exciter is an open-loop system, and the strain data are not at the same level from MFC (in Figure 5a) and PPA (in Figure 6a). For the sake of comparability between these two patches, we standardized the electric responses under the unit of strain amplitude ($1\times 10^{-3}$). For example, the strain amplitude from MFC is 1.96 $\times 10^{-3}$ in Figure 5a, and the corresponding voltage amplitude is 1.019 V in Figure 5b. Through standardization, the comparable voltage is standardized to 0.52 V (1.019 V/1.96 V) amplitude. At the same time, the comparable current/power is standardized to 6.23 μA/6.9 μW. For PPA, the comparable voltage, current, and power are 14.85 V, 1.50 mA, and 12.83 mW, respectively. By contrast, it is obvious that the harvesting capability of PPA is much more superior as compared with that of MFC, where voltage/current collection from the PPA is 28/241 times larger than MFC, and amazingly, power collection is 1986 times larger than that of MFC.

#### 2.3.3. Harvesting Characteristic Investigation

In addition to the electric harvesting capacity, we further investigate their electromechanical characteristics, which is pivotal to explore the energy conversion mechanism. As shown in Figure 7a,c, it is quite clear that the voltage/current-strain curve is in the rotational symmetric shape of ellipses suggesting their hysteretic nonlinear relationship. In addition, the power-strain curve is axial symmetry and it is quite unique in that it looks like a butterfly shape, as shown in Figure 7b,d. We previously discussed that a larger peak power indicated a stronger harvesting capability of a harvester. In fact, we found the fullness of the butterfly shape also indicated the harvesting capability of the harvester. For example, the power-strain butterfly in MFC, shown in Figure 7d, is much thinner than that in PPA, as shown in Figure 7c, which demonstrates the same conclusion (as proven in Section 2.3.2) that PPA has a much greater capability of energy harvesting than MFC.

#### 2.3.4. Rational Resistance Determination

Herein, we observe how the resistance load affects the harvested electric power, given that the excitation frequency is set as its approximate resonant frequency. In Figure 8, for comparison, all the generated powers of the cases are normalized within the unit range (0–1). The normalized powers of the MFC are listed with different resistances from 0 to 90 kOhm, as shown in Figure 8a. We find the electric power of the MFC almost increases linearly along with the resistant load. Unlike the MFC, the power of the PPA increases to a peak power, and then decreases, with an increase in resistance load. Take PPA without mass as example, the peak power is stimulated when resistance is changed to 10 kOhm. This value of resistance can be deemed to be a rational setting in the circuit for this case. Likewise, the rational resistances are all determined to be, respectively, 30 kOhm, 35 kOhm, and 50 kOhm in the cases of 12.5 g, 18.5 g, and 25 g levels for cantilever PPA.

#### 2.3.5. Appropriate Excitation Amplitude Determination

We also observe how the excitation amplitude load affects the generated power. Figure 9 displays the relationship of the generated peak/average power with the excitation amplitude from the PPA harvester under two typical cases. In Figure 9a for the case of 100 Hz frequency, 10 kOhm resistance without mass, the results demonstrate that a larger excitation amplitude stimulates more electric power. In addition, the power increases more dramatically when the harvester is excited under a larger amplitude of excitation. Likewise, Figure 9b shows the results of another typical case that represent similar changes in the generated electric power with the excitation amplitude.

The above phenomena illustrate that a more intense vibrational environment would arouse much more electric energy, which is preferable for harvesting more electric power converted from mechanical energy. However, one concern is that the structure and its harvester should stay safe with respect to their mechanical strength.

## 3. Coupled Electromechanical Modeling

### 3.1. Piezoelectric Basic Equation

When a poled piezoelectric material is mechanically strained, on the one hand, it becomes electrically polarized and electric charge is generated on the material’s surface. The phenomenon is known as the “direct piezoelectric effect” and is the basis on which piezoelectric materials are utilized as a sensor or harvester. If electrodes are connected to the piezoelectric surfaces, on the other hand, the generated electric charge can be gathered and used. This phenomenon is known as the “inverse piezoelectric effect”, and it is utilized especially as an actuator in vibration control applications.

The above phenomena are usually described by using a set of constitutive equations. On the assumption that the total strain in the transducer is the sum of mechanical strain induced by the mechanical stress and the controllable actuation strain caused by the applied electric voltage, their strain-charge form is given by:(1)$$S_{i}=s_{ij}^{E}\cdot T_{j}+e_{mi}\cdot E_{m}$$
(2)$$D_{m}=e_{mi}\cdot T_{i}+\varepsilon_{ik}^{T}\cdot E_{k}$$
where T is the stress vector; S is the strain vector; E is the vector of applied electric field; D is the vector of electric displacement; s is the matrix of compliance coefficients; e is the matrix of piezoelectric coupling coefficients; ε is the matrix of electric permittivity; and the indexes i, j = 1, 2, …, 6 and m, k = 1, 2, 3 refer to different directions within the material coordinate systems, as illustrated in Figure 10.

The constitutive Equation (2) describes the “direct piezoelectric effect” about how electric displacement is aroused in the stress field and electric field. Based on this, the collected electric charge can be determined from the electric displacement when the piezoelectric transducer is subject to a non-zero stress field, that is:(3)$$q=\iint \left[D_{1}\;D_{2}\;D_{3}\right]\left[\begin{array}{c} dA_{1}\\ dA_{2}\\ dA_{3} \end{array}\right]$$
where $dA_{1}$,$\;dA_{2}$, and $dA_{3}$ indicate the differential electrode areas in y-z, x-y, and x-z planes, respectively, as shown in Figure 10. Then, the aroused inner voltage $V_{p}$ can be obtained via $V_{p}=\frac{q}{C_{p}}$, where $C_{p}$ is the capacitance of the piezoelectric layer.

Combined with the piezoelectric basic equations, in this paper, we adopt a coupled electromechanical model in order to effectively describe the complicated vibration-to-electric behaviors of the laminated piezoelectric cantilever patch. Note that a piezoelectric ceramic material is commonly adopted due to its high-power density and simple structure, however, it demonstrates worse tenacious, flexible, and fatigue performance. Therefore, substructures such as some polymer and metallic materials are usually laminated with piezoelectric ceramic for improving those defects. On the basis of this, we present a suitable model for laminated piezoelectric structures.

### 3.2. Coupled Electromechanical Model

#### 3.2.1. Coupled Mechanical Equations

In terms of the mechanical problem, the configuration is connected to a resistor with resistance load R, as shown in Figure 11. Suppose the harvester is a laminated cantilever patch consisting of N piezoelectric layers and K substructure layers. In view of Euler–Bernoulli beam assumptions, the partial differential equation of the cantilever beam with a tip mass under an excitation at fixed end (displacement $y_{b}$) can be expressed by:(4)$$\frac{\partial^{2}M\left(x,t\right)}{\partial x^{2}}+c_{s}\Lambda \frac{\partial^{5}y\left(x,t\right)}{\partial x^{4}\partial t}+c_{a}\frac{\partial y\left(x,t\right)}{\partial t}+\bar{m}\frac{\partial^{2}y\left(x,t\right)}{\partial t^{2}}=-\left[\bar{m}+m\delta \left(x-l\right)\right]\frac{\partial^{2}y_{b}\left(x,t\right)}{\partial t^{2}}$$
where $c_{s}/c_{a}$ is the damping coefficient of strain rate/viscous air, Λ is the inertia moment of the cross section, $\bar{m}$ is the mass per unit length l, m is the tip mass, δ is the Dirac delta function, $y_{b}$ is the *y*-direction displacement at base position, y(x,t) is the relative deflection to its base at position x and time t, and M is the internal bending moment, i.e.,:(5)$$M\left(x,t\right)=-b\cdot \left(\overset{K}{\underset{pi=1}{\sum }}\int_{h_{pi-1}}^{h_{pi}}T_{1}^{pi}ydy+\overset{N}{\underset{sj=1}{\sum }}\int_{h_{sj-1}}^{h_{sj}}T_{1}^{sj}ydy\right)$$
where b and h are the width and the thickness of the laminated beam, respectively; $h_{pi}$ is the thickness of the pi-th piezoceramic layer and there are altogether K piezoceramic layers; $h_{si}$ is the thicknesses of the sj-th substructure layers and there are altogether N substructure layers; the axial stress components in the sj-th substructure at the 1-direction are presented by $T_{1}^{sj}$, and the pi-th piezoceramic layer at the 1-direction is presented by $T_{1}^{pi}$.

The piezoceramic part abides by constitutive equations (see Equations (1) and (2)) and the substructure parts abide by Hooke’s law in substructures. Then, the axial strain at 1-direction from neutral axis of the composite beam $S_{1}$ is proportional to the curvature of the beam at x position, that is:(6)$$S_{1}\left(x,\;y,\;t\right)=-y\frac{\partial^{2}y\left(x,t\right)}{\partial x^{2}}$$

The electric field component $E_{3}$ should be expressed in terms of the respective voltage term in each piezoelectric layer, as shown in Figure 11. It should be noted that different (series or parallel) connections of the piezoelectric layers differ from each other resulting in different mechanical equations. Herein, if the piezoelectric layers are assumed to be identical, the voltage generated from the piezoceramic layers is $V_{p}$. In the series connection, the voltage across the electrodes of each layer should be divided by K, which is equal to $V_{p}\left(t\right)/K$. For the case of parallel connection, the voltage is equal to $V_{p}\left(t\right)$. Then, the instantaneous electric field $E_{3}$ in series connection can be equally distracted, that is:(7)$$E_{3}=V_{p}\left(t\right)/\overset{K}{\underset{pi=1}{\sum }}h_{pi}$$

Likewise, for the configuration with parallel connection, the electric fields in each layer are:(8)$$E_{3}=V_{p}\left(t\right)/\left(\frac{1}{K}\cdot \overset{K}{\underset{pi=1}{\sum }}h_{pi}\right)$$
where $h_{pi}$ is the assumed center-to-center interdigitated electode spacing of the pi-th piezoelectric layer.

Combined with the relationships from bending moment, curvature to the axial strain in the beam in Equation (6), Equation (5) can be replaced by:(9)$$M\left(x,t\right)=-EI\frac{\partial^{2}y\left(x,t\right)}{\partial x^{2}}+\varsigma \cdot V_{p}\left(t\right)\left[H\left(x\right)-H\left(x-l\right)\right]$$
where EI is the bending stiffness of the composite cross section of the laminated beam, H(x) is the Heaviside function, and ς is a coupling coefficient.

On the assumption of the overall stiffness equal to a sum of that from every layer, the bending stiffness term EI is usually divided into two parts (one refers to the piezoelectric layers, and the other refers to the substructural layers). It is given by:(10)$$EI=\overset{K}{\underset{pi=1}{\sum }}c_{11}^{E}I_{pi}+\overset{N}{\underset{sj=1}{\sum }}E_{sj}I_{sj}$$
where $E_{sj}$ is the Young modulus of sj-th substructure layer, $I_{sj}$ are the relative substructural inertia moments of the cross section, and $c_{11}^{E}$ is the elastic stiffness component at constant electric field.

In Equation (9), the coupling coefficient ς can be expressed by:(11)$$\;\varsigma =e_{31}b\cdot \vartheta \left(h_{pi},h_{si}\right)$$
where the coupling geometric coefficient ϑ is dependent on the dimensional and combinatorial relationship of each substructure and piezoelectric layers ($h_{pi}$ and $h_{si}$). It is also dependent on the circuit connection method.

Based on the above, the coupled mechanical Equation (4) can be substituted with:(12)$$EI\frac{\partial^{4}y\left(x,t\right)}{\partial x^{4}}+c_{s}I\frac{\partial^{5}y\left(x,t\right)}{\partial x^{4}\partial t}+c_{a}\frac{\partial y\left(x,t\right)}{\partial t}+\bar{m}\frac{\partial^{2}y\left(x,t\right)}{\partial t^{2}}-\varsigma v_{p}\left(t\right)\left[\frac{d\delta \left(x\right)}{dx}-\frac{d\delta \left(x-l\right)}{dx}\right]=-\left[\bar{m}+m\delta \left(x-l\right)\right]\frac{\partial^{2}y_{b}\left(x,t\right)}{\partial t^{2}}$$

Equation (4) is a partial differential equation and its analytical solution is difficult, and even impossible, to solve. Herein, we employ the mode superposition method to solve it effectively. The model expression is as follows:(13)$$y\left(x,t\right)=\overset{\infty }{\underset{k=1}{\sum }}\varphi_{k}\left(x\right)\cdot q_{k}\left(t\right)$$
where $q_{k}\left(t\right)$ is the modal mechanical response, and $\varphi_{k}\left(x\right)$ is the mass normalized eigenfunction of the *k*-th vibration mode whose expression is:(14)$$\varphi_{k}\left(x\right)=C_{k}\left[\cos h\frac{\beta_{k}x}{L}-\cos\frac{\beta_{k}x}{L}-\frac{\sin\beta_{k}-\sin h\beta_{k}+\beta_{k}\frac{m}{\bar{m}L}\left(\cos\beta_{k}-\cos h\beta_{k}\right)}{\cos\beta_{k}+\cos h\beta_{k}-\beta_{k}\frac{m}{\bar{m}L}\left(\sin\beta_{k}-\sin h\beta_{k}\right)}\cdot \left(\sin h\frac{\beta_{k}x}{L}-\sin\beta_{k}\right)\right]$$
where $C_{k}$ is a modal amplitude constant that can be evaluated by normalizing the eigenfunctions according to the orthogonality conditions, and $\beta_{k}$ is the eigenvalues for k mode that can also be evaluated by the short circuit conditions [22].

By adopting the mode superposition method, Equation (12) is replaced by the mechanical equation of motion in modal coordinates, that is:(15)$$\frac{d^{2}q_{k}\left(t\right)}{dt^{2}}+2\xi_{k}\omega_{k}\frac{dq_{k}\left(t\right)}{dt}+\omega_{k}^{2}q_{k}\left(t\right)-\lambda_{k}v_{p}\left(t\right)=f_{k}\left(t\right)$$
where $\omega_{k}$ is the undamped natural frequency of the kth vibration mode, $\xi_{k}$ is the modal mechanical damping ratio derived from the proportional damping assumption. The modal electromechanical coupling term $\lambda_{k}={\varsigma \frac{d\varphi_{k}\left(x\right)}{dx}|}_{x=L}$ and modal mechanical forcing function is:(16)$$f_{k}\left(t\right)=-\bar{m}\frac{d^{2}y_{b}\left(t\right)}{dt^{2}}\int_{0}^{L}\varphi_{k}\left(x\right)dx-m\varphi_{k}\left(l\right)\frac{d^{2}y_{b}\left(t\right)}{dt^{2}}$$

#### 3.2.2. Coupled Electric Equations

In the case of a laminated cantilever, the mechanical strain can be assumed to be the axial strain in the case of curvature. Then, the tensorial expression of the piezoelectric constitutive (Equations (1) and (2)) can be simplified to scalar equations as:(17)$$T_{1}=c_{11}^{E}S_{1}-e_{31}E_{3}$$
(18)$$D_{3}=e_{31}S_{1}+\varepsilon_{33}^{s}E_{3}$$
where $c_{11}^{E}$ is the elastic stiffness of the piezoelectric layer at constant electric field, $e_{31}$ is the piezoelectric constant, and $\varepsilon_{33}^{s}$ is the permittivity component at constant strain with the plane-stress assumption. Note the subscripts 1, 2, and 3 represent longitudinal, lateral, and vertical direction of the cantilever beam, respectively, as shown in Figure 11.

The axial strain components of piezoelectric derived from Equation (17) and substructure layers in the laminated patch is proportional to its curvature at the position *x*. Based on this, the stimulated electric field $E_{3}$ can be solved.

Because the circuit admittance across the electrodes is 1/R, the electric response is obtained based on Gauss’s law in Equation (3) as:(19)$$\frac{V_{p}\left(t\right)}{R}=\frac{d}{dt}\left(\int_{A}D_{3}dA_{3}\right)$$
where $D_{3}$ is the 3-directional electric displacement in the piezoelectric layer and $A_{3}$ is the electrode area in the x-z plane.

Due to the average strain in the piezoelectric layer subjected by the bending, substitute Equation (20) into Equation (21), and then yield:(20)$$-e_{31}h_{pc}b\int_{0}^{L}\frac{\partial^{3}y\left(x,t\right)}{\partial x^{2}\partial t}dx=C_{p}\;\frac{dV_{p}\left(t\right)}{dt}+\frac{V_{p}\left(t\right)}{R}$$
where $C_{p}$ is capacitance equal to $\frac{bl\varepsilon_{33}^{S}}{h_{p}}$ in the electric circuit, and $h_{pc}$ is the equivalent distance between the neutral axis and the center of piezoceramic layer.

Based on Kirchhoff law, the current source $I_{p}$ can be derived as:(21)$$I_{p}\left(t\right)=C_{p}\frac{dV_{p}\left(t\right)}{dt}+\frac{V_{p}\left(t\right)}{R}$$

It is easy to observe that Equations (20) and (21) have the same terms on the right side. Thus, current $I_{p}$ can be deduced directly by the agreement with the left term between them.

In fact, the measured electric results in the experiment are received from the electric circuit through data acquisition, not the generated ones ($I_{p}$ and $V_{p}$) directly from the harvester. It is easy to understand the voltage across the resistance should be the same as the generated one, i.e., $V\left(t\right)=V_{p}\left(t\right)$, because they have a parallel relationship. Therefore, the measured current I(t) from the experiment can be calculated based on Ohm’s Law shown below as:(22)$$I\left(t\right)=V_{p}\left(t\right)/R$$

Next, substitute the modal expansion to Equation (20), and then achieve:(23)$$\overset{\infty }{\underset{k=1}{\sum }}\theta_{k}\frac{dq_{k}\left(t\right)}{dt}=C_{p}\cdot \frac{dV_{p}\left(t\right)}{dt}+\frac{V_{p}\left(t\right)}{R}$$
where the modal coupling term $\theta_{k}=-e_{31}h_{pc}b\int_{0}^{l}\frac{d\varphi_{k}\left(x\right)}{dt}dx$.

#### 3.2.3. Coupled Electromechanical Equations

The mechanical equation (Equation (15)), and electric circuit equation (Equation (23)), present coupled mechanical and electric relations of the variables modal mechanical response $q_{k}\left(t\right)$ and voltage response V(t) or $V_{p}\left(t\right)$. Based on them, the variables $q_{k}\left(t\right)$ and V(t)/$V_{p}\left(t\right)$ can be theoretically calculated. Moreover, the electric circuit equations (Equations (21) and (23)), are combined to calculate the results of source current $I_{p}\left(t\right)$ and measured current I(t), successively.

### 3.3. Coupled Equations’ Solutions

To derive a steady-state solution from the coupled electromechanical equations, we employ frequency response functions (FRFs) to solve their mechanical and electrical results. On the assumption that the 3rd directional displacement ($y_{b}$) in Equation (4) is harmonic, its modal forcing function can be expressed as $f_{k}\left(t\right)=F_{k}e^{\omega tj}$. Derived from Equation (16), amplitude $F_{k}$ can be then calculated as:(24)$$F_{k}=\omega^{2}\left[\bar{m}Y_{0}\int_{0}^{L}\varphi_{k}\left(x\right)dx+m\varphi_{k}\left(l\right)\right]$$
where $Y_{0}$ represents the deformation amplitude at the 3rd direction from the base, ω represents the excitation frequency, and j represents the imaginary part numeric values.

In this paper, model mechanical $q_{k}\left(t\right)=Q_{k}e^{\omega tj}$ and voltage response $V\left(t\right)=V_{0}e^{\omega tj}$ are deemed to be single frequency harmonic whose amplitudes $Q_{k}$ and $V_{0}$ are complex valued based on the hypothesis of linear system. Therefore, the solution of the model (Equations (20) and (23)) can be replaced as an equation set:(25)$$\lambda_{k}V_{0}+\left(\omega_{k}^{2}-\omega^{2}+2\xi_{k}\omega_{k}\omega \cdot j\right)Q_{k}=F_{k}$$
(26)$$\left(\frac{1}{R}+\frac{1}{2}C_{p}\omega \cdot j\right)V_{0}-\omega \overset{\infty }{\underset{k=1}{\sum }}\theta_{k}Q_{k}\cdot j=0$$

As soon as $Q_{k}$ and $V_{0}$ are determined based on the above, the mechanical and electric solutions are obtained. The presentational electric expressions are listed below as:(27)$$V\left(t\right)=e^{\omega tj}\cdot \frac{\sum_{k=1}^{\infty }\frac{\omega \theta_{k}F_{k}\cdot j}{\omega_{k}^{2}-\omega^{2}+2\xi_{k}\omega_{k}\omega \cdot j}}{\frac{1}{R}+\frac{1}{2}C_{p}\omega \cdot j+\sum_{k=1}^{\infty }\frac{\omega \theta_{k}F_{k}\cdot j}{\omega_{k}^{2}-\omega^{2}+2\xi_{k}\omega_{k}\omega \cdot j}}$$
(28)$$I\left(t\right)=e^{\omega tj}\cdot \frac{\sum_{k=1}^{\infty }\frac{\omega \theta_{k}F_{k}\cdot j}{\omega_{k}^{2}-\omega^{2}+2\xi_{k}\omega_{k}\omega \cdot j}}{1+\frac{1}{2}C_{p}\omega R\cdot j+\sum_{k=1}^{\infty }\frac{\omega \theta_{k}F_{k}\cdot j}{\omega_{k}^{2}-\omega^{2}+2\xi_{k}\omega_{k}\omega \cdot j}R}$$

The complex voltage amplitude $V_{0}$ can be substituted into Equation (27) to achieve the steady mechanical response expression as follows:(29)$$\eta \left(t\right)=\left(F_{r}-\lambda_{k}\frac{\sum_{k=1}^{\infty }\frac{\omega \theta_{k}F_{k}\cdot j}{\omega_{k}^{2}-\omega^{2}+2\xi_{k}\omega_{k}\omega \cdot j}}{\frac{1}{R}+\frac{1}{2}C_{p}\omega \cdot j+\sum_{k=1}^{\infty }\frac{\omega \theta_{k}F_{k}\cdot j}{\omega_{k}^{2}-\omega^{2}+2\xi_{k}\omega_{k}\omega \cdot j}}\right)\times \frac{e^{\omega tj}}{\omega_{k}^{2}-\omega^{2}+2\xi_{k}\omega_{k}\omega \cdot j}$$

Then, the transverse deformation response at point x can be achieved by substituting Equation (29) into Equation (13) as:(30)$$y\left(x,t\right)=\overset{\infty }{\underset{k=1}{\sum }}\left[\left(F_{r}-\lambda_{k}\frac{\sum_{k=1}^{\infty }\frac{\omega \theta_{k}F_{k}\cdot j}{\omega_{k}^{2}-\omega^{2}+2\xi_{k}\omega_{k}\omega \cdot j}}{\frac{1}{R}+\frac{1}{2}C_{p}\omega \cdot j+\sum_{k=1}^{\infty }\frac{\omega \theta_{k}F_{k}\cdot j}{\omega_{k}^{2}-\omega^{2}+2\xi_{k}\omega_{k}\omega \cdot j}}\right)\times \frac{e^{\omega tj}}{\omega_{k}^{2}-\omega^{2}+2\xi_{k}\omega_{k}\omega \cdot j}\right]+y_{b}\left(x,t\right)$$

## 4. Coupled Electromechanical Modeling by Using Identified Parameters

### 4.1. Model Parameters

In the coupled electromechanical model, there are many parameters that must be assigned accurate values, otherwise the model would demonstrate inaccurate, even invalid, results. In fact, it is quite challenging, if not impossible, to determine accurate values for all these parameters. This could be attributed to the following three reasons: One reason is that some parameters are difficult to clearly defined, and thereby, they are often simplified based on sets of certain assumptions. The simplification easily causes the model to provide inaccurate results. The second reason is that most laminated harvesting products are encapsulated, and the configuration, geometrical, mechanical, etc. information for every layer is not very specific. The third reason is that the design of some laminated harvesters is complicated, such as MFC, in which the “piezoelectric layer” is not made of a single piezoelectric material but fibers that are embedded in a polymer matrix.

In view of the above, we propose one strategy to deal with the crucial issues. First, we outline which parameters are not specifically defined and are difficult to assign in the model.

(1)Parameter $h_{pi}$ represents the center-to-center interdigitated electrode spacing (assumed as the thickness) of the pi-th piezoelectric layer. It is simplified on the assumption that identical and electric fields at each layer are equally distracted, which may cause the model’s result to be not accurate enough.(2)Parameter $h_{pc}$ represents (equivalent) the distance from the neutral axis of the structural cross section to the center of the piezoelectric layer. Similar to $h_{pi}$, it is simplified on the assumptions that identical electric fields at each layer are equally distracted. In addition, it could be even harder to assign when the harvester has multiple piezoelectric layers, and how to propose its proper equivalent value is another issue.(3)Parameter EI represents the equivalent bending stiffness of the composite cross section for the constant electric field condition. In order to obtain its value, simplified relations (referred in Equation (10)) are commonly proposed based on some inevitable assumptions by ignoring non-uniformity field and coupling effect within layers.(4)Parameter ϑ represents the coupling geometric coefficient that is comprehensively dependent on the thickness of every layer in the harvester, and also dependents on the circuit connection method. In general, it is a complicated coefficient that is not direct and easy to obtain accurately.

It takes little effort to discover that the above parameters have comprehensively equivalent definitions, and it is quite difficult to assign them accurate values. The simplification of hypothetical principles also produces more difficulties resulting in inaccurate, or even invalid, evaluations of them. Thus, determining effective values for the abovementioned parameters is pivotal for determining the model’s validity. In this paper, we set out to propose a method of parameter identification for those four parameters by using a genetic algorithm (GA) procedure, which is introduced in the next subsections.

### 4.2. Parameter Identification Procedure

Taking the example of a laminated harvester with a single piezoelectric layer, there are four parameters to identify which are $h_{pc}$, $h_{p1}$, EI, and ϑ. We propose Equation (31) with these four parameters (instead of electromechanical expressions (Equations (15), (21) and (23)) to describe the mechanical-electric behavior in a cantilever harvester. It is given by:(31)$${\left[V,I\right]}_{sim}=f\left(S_{1},\;\tilde{\Theta }\;\right)$$
where Θ denotes the vectors of the identified parameters ($h_{pc},\;h_{p1},\;EI,\;\mathrm{and}\;\vartheta$) that represent the intrinsic attributes of the harvesting systems. In order to determine the vector Θ, a unified optimization strategy can be qualified through measuring the discrepancy between model predictions and experimental data. The smaller the discrepancy, the closer the identified parameters $\;\tilde{\Theta }$ is to the intrinsic ones Θ. Herein, we employ a genetic algorithm (GA) to carry out the said minimization due to its powerful computational capability of multi-agent parallelized stochastic search.

A flow chart on the GA for parameter estimation is shown in Figure 12. First, experimental data were collected, which included the mechanical data (strain $S_{1,exp}$) and electric data (voltage $V_{exp}$ and current $I_{exp}$). Secondly, we initialized a set of parameters $\;\tilde{\Theta }$ and incorporated the strain data $S_{1,exp}$ into the model. Then, the simulated voltage $V_{sim}$ and current $I_{sim}$ were calculated. During the phase of the GA, the fitness function “Fit” is defined as the discrepancy between the experimental data and simulated result from the model. The lower the fitness function value, the better the individual chromosome. Through appropriate iterative circulation, the optimal individual (identified parameters) is eventually computed out. Note that the above variables with subscript “sim/exp” indicate simulation/experimental data.

### 4.3. Genetic Algorithm (GA) Detail

A GA involves several basic working parts including chromosome representation, fitness function, genetic operators, and termination criteria.

(1)Chromosome representation

In GA terminology, chromosomes are simply the variations in the set of unknown variables of the problem that the GA is going to solve. In this study, the model tackled has four variables that are represented simply by Θ. Thus, the chromosome representation is expressed as:(32)$$\Theta_{u}=\left\{h_{pc,u\;},\;h_{p1,u},\;EI_{u},\;\vartheta_{u}\;\right\},\;u=1,\ldots ,\;U$$
where u denotes the u-th individual in a chromosome set, whose total number is U.

(2)Fitness function

A fitness function measures the status of one chromosome achieving the preset objective. Each chromosome needs to be awarded an evaluation that indicates how close it came to meeting the objective. Here, we define the fitness function (“Fit”) by using a combination weight function including the root-mean-square error (*RMSE*) for each output (voltage and current responses) that is designed between the simulated and experimental results, which are:(33)RMSEV,u=1W{∑w=1W[Vexp,w(S,Θ)−Vsim,w(S,Θ˜)]2}1W[∑w=1WVexp,w(S,Θ)]2
(34)RMSEI,u=1W{∑w=1W[Iexp,w(S,Θ)−Isim,w(S,Θ˜)]2}1W[∑w=1WIexp,w(S,Θ)]2
(35)$$\mathrm{Fit}=\alpha \cdot RMSE_{V,min}+\left(1-\alpha \right)\cdot RMSE_{I,min}$$
where $RMSE_{V,u}$/$RMSE_{I,u}$ denotes the voltage/current’s discrepancy between simulation and experimental results at the u-th individual. $V_{exp,w}\;$and $V_{sim,w}$ denote the voltage, respectively, from experimental and simulation results (as subscribed with “exp” and “sim”) at the w-th time point whose total number is W. The fitness function consists of voltage and current discrepancies. A weight coefficient “α” is also employed for balancing the importance among voltage and current predictions, and then yields the fitness function in Equation (35).

(3)Termination strategy

The traditional GA empirically adopts a fixed number of generations, which either lead to excessive computations or solutions with inadequate accuracy. To avoid this issue, a set of termination rules is designed on the aspects of global accuracy and average accuracy. The rules can comprehensively guarantee the best and the average individuals of every generation resulting in sufficient accurate solutions with appropriate computation loading. The rules are set as follows:

**Rule 1:** Define a minimum threshold $e_{1,min}$. As soon as the finest fitness value meets $e_{1,min}$ at the v-th iteration, automatically terminate the procedure and extract the best solution, that is:(36)$${\mathrm{Fit}}_{v}\leq e_{1,min}$$
where $e_{1,min}$ is empirically set as 10% in the present study.

**Rule 2:** Define a calibrated threshold $e_{2,min}$. As soon as the difference of the finest fitness values between the kth iteration and (v+1)-th iteration meets the $e_{2,min}$, automatically terminate the procedure and extract the best solution. It is given by:(37)$$\left|{\mathrm{Fit}}_{v}-{\mathrm{Fit}}_{v+1}\right|\leq e_{2,min}$$
where $e_{2,min}$ is empirically set as 10% of the $e_{1,min}$ in the present study.

In general, Rule 1 is responsible for guaranteeing the accuracy of the final solution, as well as Rule 2 is responsible for avoiding excessive and inefficient computation.

Other than the above, there are some other basic settings needed to complete the GA procedure. Here, the population is set as 60 individuals with 20 binary genes, the roulette wheel selection with elitism strategy is used, and the crossover/mutation operator is set as 0.85/0.01.

## 5. Model Validation with Experimental Data

In this study, we introduce a coupled electromechanical model with a set of identified parameters that are obtained by using GA procedure. In this section, we verify the validity of the proposed method.

### 5.1. Comparison of Model and Experimental Data

#### 5.1.1. Original Model (without Original Parameters)

Based on related simplified equations, the parameters are evaluated, referred to as the original parameters in this study.

Then, taking PPA for example, the original parameters are deduced and evaluated as follows:

(1) Parameter $h_{p1}$ is the equivalent thickness of the piezoelectric layer and is equal to 50 μm. (2) Parameter $h_{pc}$ is the distance from the center of the piezoelectric layer to the structural neutral axis. Then, the relation is deduced as $h_{pc}=\left|h_{s4}+h_{s3}-h_{s2}-h_{s1}\right|/2$ and is equal to 150 μm. (3) Parameter EI is deduced based on Equation (10) as $EI=\overset{4}{\underset{sj=1}{\sum }}E_{s1}\cdot I_{s1}+c_{11}^{E}\cdot I_{p}$ and is equal to 107 $N\cdot {\mathrm{cm}}^{2}$. (4) Parameter ϑ is dependent on the thickness of every layer and the circuit connection method. It is deduced as $\vartheta =\left|h_{s4}+h_{s3}-h_{s2}-h_{s1}\right|$ and is equal to 10 mm. The valuations are listed in Table 1.

By using the model with original parameters, representative results (under the case of 120 Hz excitation, 10 kOhm resistance, and 0 g tip mass) from the PPA harvester are displayed in Figure 13. It is easy to observe that the input strain (see Figure 13a) is nearly the same between the model simulation and experimental data in terms of the sinusoidal amplitude (around 0.56 $\times 10^{-3}$) and frequency (exactly 120 Hz). After substituting the strain data into the model, we observe the mechanical response (tip deformation of y direction), which represents the exact same frequency as the experimental data and an amplitude around 1.6 mm that is close to the experimental data, as illustrated in Figure 13b. Next, the electric response (current and voltage) is slightly smaller, around 12%, as compared with the experimental data, as shown in Figure 13b,c. For the comparison of characteristic relationships, the strain-voltage characteristic loops are illustrated; the elliptical figures in simulation are slightly rotated anticlockwise versus those from the experimental data, as shown in Figure 13e. The strain-power characteristic loops are shown in Figure 13f; the butterfly shapes in simulation are a vertically compressed version of those in experimental mode. Meanwhile, the maximum power prediction from the model simulation is much smaller, around 30%, as compared with that of the experimental data.

Next, taking MFC for example, we find that its piezoelectric layer is assembled of piezoelectric fibers with polymer matrix and interdigited electrodes, and it is impossible to obtain the parameters’ values by using conventional methods.

#### 5.1.2. Proposed Model (with Updated Parameters)

Using the GA procedure, we identified the parameter values which are listed in Table 2. Then, we substituted the updated parameter values into the model again. The mechanical responses (strain and deformation response) using the updated parameter values in the model are basically the same as the experimental data, shown in Figure 13a,b. However, the electric responses (current and voltage) have better agreement with the experiment data, shown in Figure 14a,b. For The comparison of characteristic relationships showed that the strain-voltage loops between simulation and experiment data were even more identical. The strain-power characteristic loops, as shown in Figure 13d, are butterfly shaped in simulation and just slightly vertically compressed as compared with those in the experimental results. The maximum power prediction from the model simulation is slightly lower, around 13.9%, than that in the experimental data. Apparently, the prediction is much more accurate when the model is employed with identified parameters using the GA procedure.

According to an MFC harvester, the parameters in the model are not known because of its complicated laminated structure. Given this, we use the GA procedure to determine its potential vague parameters. Here, dataset of one random case (60 Hz excitation and 90 kOhm resistance) is utilized for the GA procedure, and its identified parameters are obtained and listed in Table 1.

Then, we can substitute the parameters into the model to predict the harvesting behavior for other cases. Figure 15 illustrates one of the simulation results that are in a good agreement with the experimental results, which proves the proposed method is efficient and accurate.

### 5.2. Error Analysis

“Fit” (from Equation (35)) is the error between the simulation and experiment results, which represents the model’s accuracy for a certain case (under a certain frequency, resistance, and mass weight). We found that the Fit values varied according to different cases. In order to test the effectiveness of the proposed model, an error analysis should be carried out. Four groups of cases in PPA, as previously mentioned in Section 2, were classified according to four weight levels at the tip (0 g, 12.5 g, 18.75 g, and 25 g). Each group consisted of cases with different excitation frequencies, while resistances were set as the optimal resistance.

Figure 16 shows the “Fit” values vs. excitation frequency for the four groups. As displayed in Figure 16a, we found the “Fit” value for the original model is lowest at 34% when the excitation frequency approaches the resonant frequency (120 Hz). When the excitation frequency is changed to a value that is farther away from the resonant frequency, the error increases further until the ultimate error of 82%. In contrast, the “Fit” values for the model with updated parameters are all reduced with a minimum of 14% and a maximum of 49%. Figure 16b,d illustrate the results of the other three groups, in the same manner. The “Fit” values for the model with updated parameters are all reduced from the values of the original model, which proves the updated model with optimized parameters is superior.

In general, electric power prediction is our most important goal. From this perspective, the simulation error should also consider the proportion of harvested power in different cases. As proven before, the harvested power is significantly dominant around the case of excitation frequency at resonance, where the model is quite accurate. In contrast, negligible harvesting energy might not matter if the model is not sufficiently accurate. In view of this, the “Fit” value cannot indicate authentic effectiveness of one modeling method for predicting generated power. It, in fact, should also consider the harvested energy percentage of the whole power among the group cases, which is called the “energy ratio (ER)”. It is given by:(38)$$\mathrm{ER}\left(r\right)=\frac{P\left(r\right)}{\sum_{r=1}^{N}P\left(r\right)}\times 100\%$$
where P(r) indicates the harvested power from the r-th (ordinal number) case, and $\overset{N}{\underset{r=1}{\sum }}P\left(r\right)$ indicates the gross power summed in each group where N is the max-ordinal number. ER represents harvesting capacities when exaction frequencies are varied.

Then, another measurable indicator called the equivalent error (EE), within each group of cases, is devised as:(39)$$\mathrm{EE}=\left\{\overset{N}{\underset{r=1}{\sum }}\left[\mathrm{ER}\left(r\right)\cdot \mathrm{Fit}\left(r\right)\right]\right\}\times 100\%$$
where Fit(r) is from Equation (35), and ER(r) is from Equation (38).

Then, as shown in Table 3, it is as expected, the “EE” from the original method is also reduced significantly when the proposed method is use. Regarding the No. 1 group of examples, simulation accuracy by using the identified parameters is remarkably improved by 56.36%. Regarding the No. 2 group, the “EE” is improved by 48.17%. The other groups show that the simulation using the proposed method is more accurate. It is obvious that the model with optimized parameters gains much better simulations for imitating the characteristic loops of a cantilever harvester owing to its full consideration of the electric energy.

### 5.3. Change Analysis of Responses with Vibration Frequency

The proposed model was validated as effective and sufficiently accurate, as described in Section 5.2. Here, we combine the experimental results and the simulation results and discuss the capacities of mechanical responses and electric responses. An example of the PPA cantilever harvester with 0 g mass under 10 kOhm resistance is shown in Figure 16, which illustrates the relationships of different responses’ amplitudes with different vibration frequencies.

In Figure 17a, the strain amplitude in the experimental results reaches a peak of 0.547 $\times 10^{-6}$ at 120 Hz vibration frequency, which closely approaches the dominant frequency (123 Hz). Under the same conditions, the predicted strain amplitude by model simulation is 0.556 $\times 10^{-6}$. As the vibration frequency changes farther away from the dominant frequency, the strain amplitude decreases rapidly until the minimum (approaching zero). In general, all the simulations for strain response are very close to the experimental results, with an error of 5.6% on average.

In Figure 17b, the deformation amplitude of the y-direction at the free tip of the cantilever also reaches a peak (1.66 mm) at a vibration frequency of 120 Hz. It also decreases rapidly towards zero when the vibration frequency changes farther from the dominant frequency. The simulations for deformation also agree well with the experimental results, with an error of 11.3% on average.

In Figure 17c, the electric voltage amplitude demonstrates a similar pattern to the vibration frequency as compared with mechanical responses. The simulation results are basically similar to the experimental results, with an average error increased to 20%.

In Figure 17d, harvested power illustrates a significant peak of 6.787 mW at 120 Hz vibration, and it decreases very sharply, approaching zero farther away from the dominate frequency. The predictions for gross power by the model are basically in good agreement with the test results within an error of 25%.

In general, mechanical responses and electric responses of cantilever harvesters all represent a very similar pattern that a resonance phenomenon triggers more easily when vibration frequency changes closer to the structural dominant frequency. On the contrary, the responses decrease rapidly approaching zero when vibration frequency changes farther from the dominant frequency. Regarding the simulation effect of the proposed model, we found that the mechanical responses are predicted quite accurately, and the electric responses are still predicted effectively, with an error that remains below 25%.

## 6. Conclusions

In this study, we investigated a vibration-based cantilever beam from two popular piezoelectric laminated patches (MFC and PPA) in terms of an experimental study of its characteristic behaviors and a modeling study of energy harvesting. In the experiment, we investigated the harvesting capacity of the system as well as the electromechanical behavior of the cantilever harvester. Furthermore, some pivotal rules with regard to several design variables were summarized, which should be very useful for maximizing energy conversion. On the basis of identified parameters, a coupled electromechanical model for a laminated piezoelectric cantilever harvester was proposed with a set of optimized parameters determined by an optimization program. The model was tested and found to be superior for accurately estimating harvested energy amount, and describing the characteristic behavior effectively. The model was also demonstrated to be more versatile and applicable for diversified harvesters with vague or unknown parameters, which is an impossible task for traditional modeling methods.

In summary, this study conducted a characteristic experiment and harvesting modeling. The combined results should support realizing more efficient conversion of vibration energy to electrical energy, and should have great application potential in piezoelectric cantilever harvesters.

## Figures and Tables

**Figure 1 sensors-22-04457-f001:**
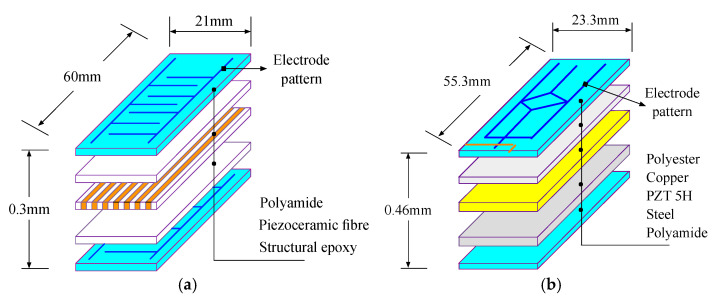
Schematic diagram for the construction of the piezoelectric laminated products. (**a**) MFC patch; (**b**) PPA patch.

**Figure 2 sensors-22-04457-f002:**
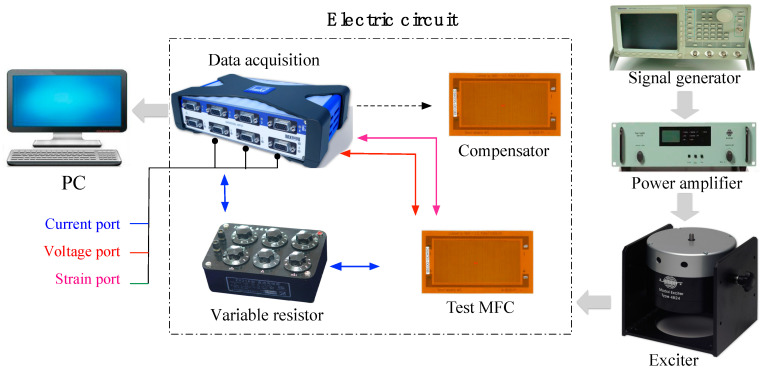
Flowchart for the energy harvesting system with MFC cantilever patch.

**Figure 3 sensors-22-04457-f003:**
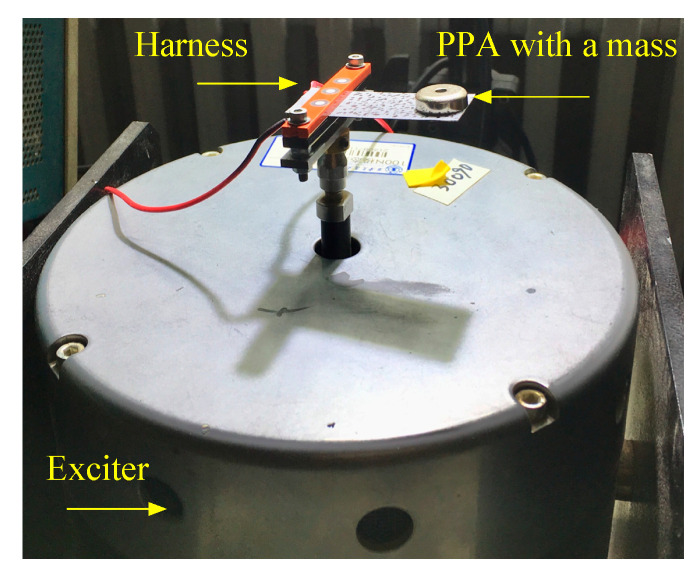
PPA clamped to the exciter.

**Figure 4 sensors-22-04457-f004:**
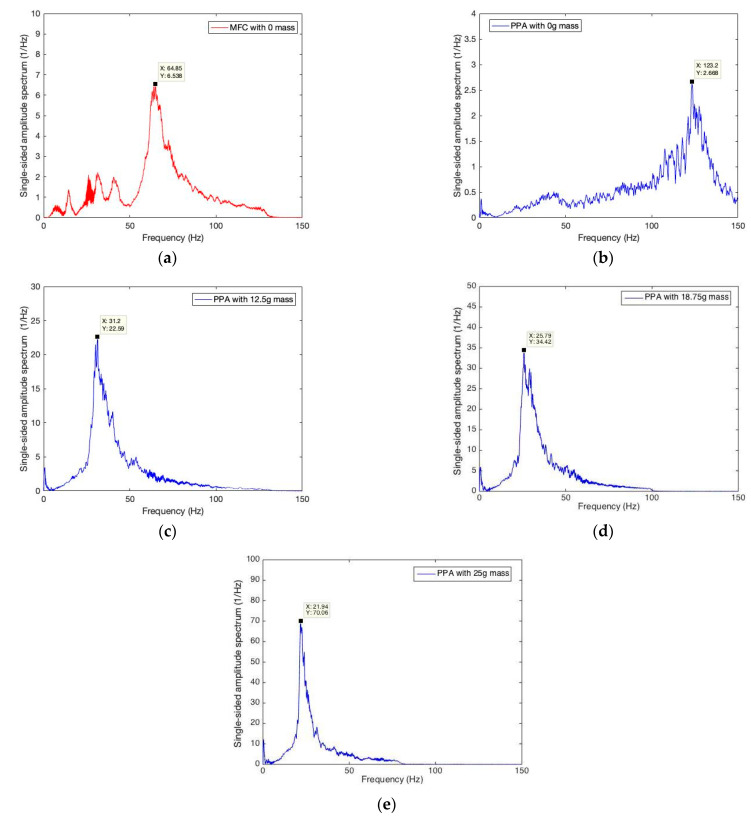
Amplitude spectrum of frequency sweep for the strain response of PPA and MFC. (**a**) Amplitude spectrum for MFC with 0 g mass; (**b**) Amplitude spectrum for PPA with 0 g mass; (**c**) Amplitude spectrum for PPA with 12.5 g mass; (**d**) Amplitude spectrum for PPA with 18.75 g mass; (**e**) Amplitude spectrum for PPA with 25 g mass.

**Figure 5 sensors-22-04457-f005:**
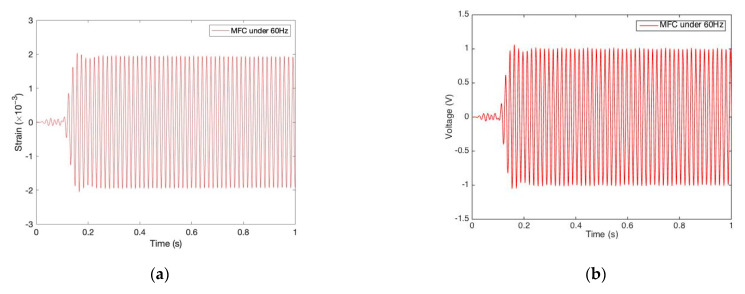
Mechanical and electric responses from MFC under resonant excitation. (**a**) Strain response for MFC under 60 Hz; (**b**) Voltage response for MFC under 60 Hz; (**c**) Current response for MFC under 60 Hz; (**d**) Power response for MFC under 60 Hz.

**Figure 6 sensors-22-04457-f006:**
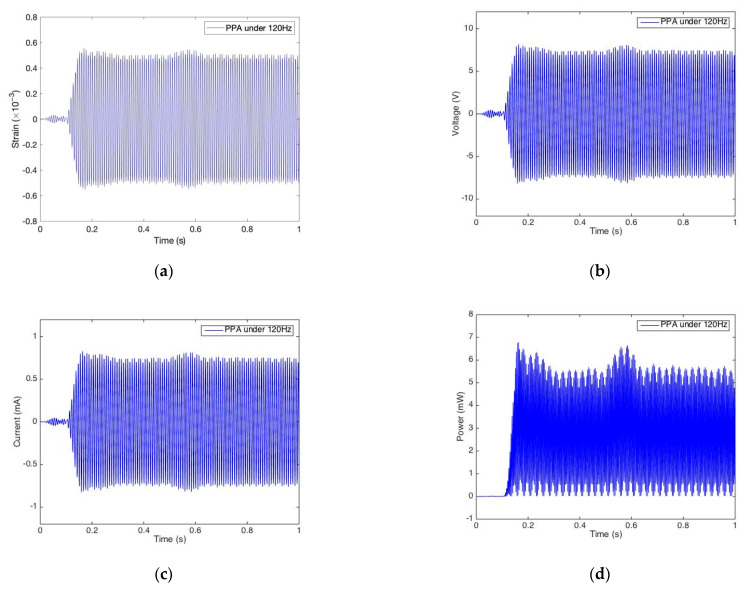
Mechanical and electric responses from PPA under resonant excitation. (**a**) Strain response for PPA under 120 Hz; (**b**) Voltage response for PPA under 120 Hz; (**c**) Current response for PPA under 120 Hz; (**d**) Power response for PPA under 120 Hz.

**Figure 7 sensors-22-04457-f007:**
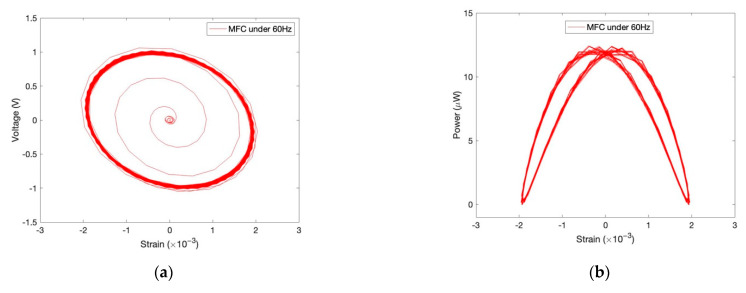
Electromechanical characteristics from MFC and PPA under resonant excitations. (**a**) Voltage vs. strain for MFC under 60 Hz; (**b**) Power vs. strain for MFC under 60 Hz; (**c**) Voltage vs. strain for PPA under 120 Hz; (**d**) Power vs. strain for PPA under 120 Hz.

**Figure 8 sensors-22-04457-f008:**
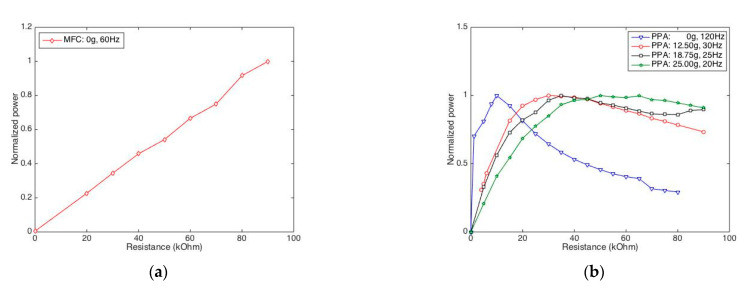
Normalized-harvested powers by varying load resistances for MFC and PPA. (**a**) Power vs. resistance for MFC; (**b**) Power vs. resistance for PPA.

**Figure 9 sensors-22-04457-f009:**
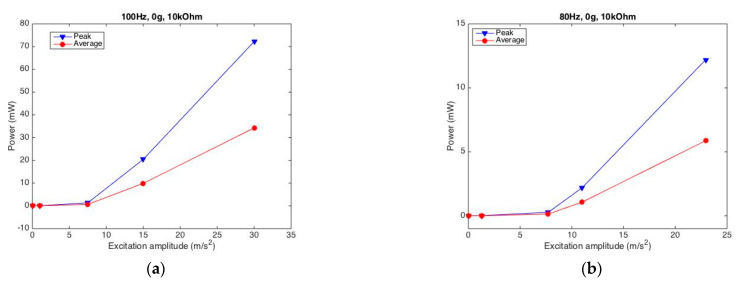
Normalized-harvested powers by varying excitation amplitudes for PPA. (**a**) under 100 Hz excitation; (**b**) under 80 Hz excitation.

**Figure 10 sensors-22-04457-f010:**
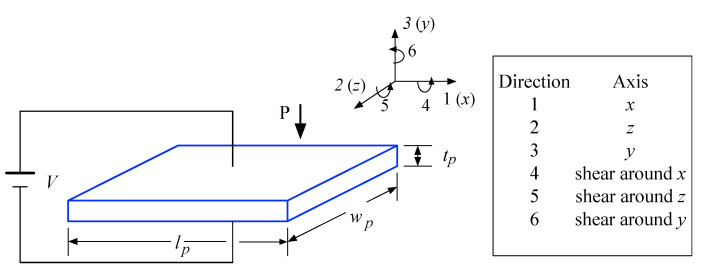
Axis nomenclature for the piezoelectric constitutive equation.

**Figure 11 sensors-22-04457-f011:**
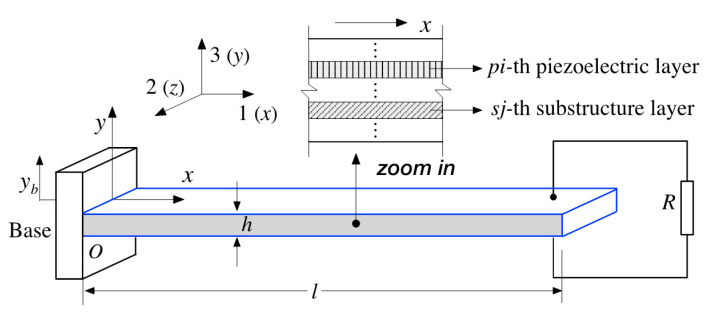
Schematic diagram of the universal composite-laminated piezoelectric cantilever harvester.

**Figure 12 sensors-22-04457-f012:**
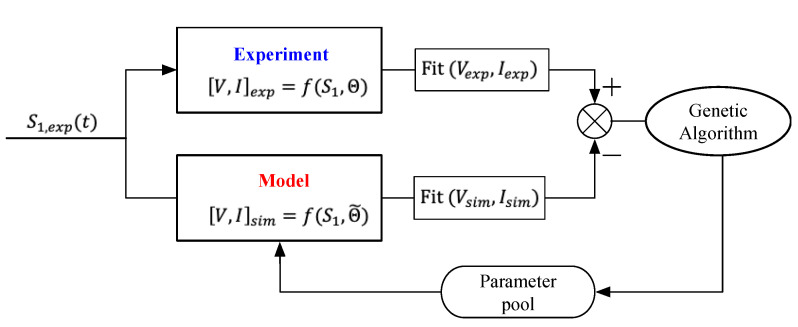
GA flowchart of parameter estimation for the electromechanical model.

**Figure 13 sensors-22-04457-f013:**
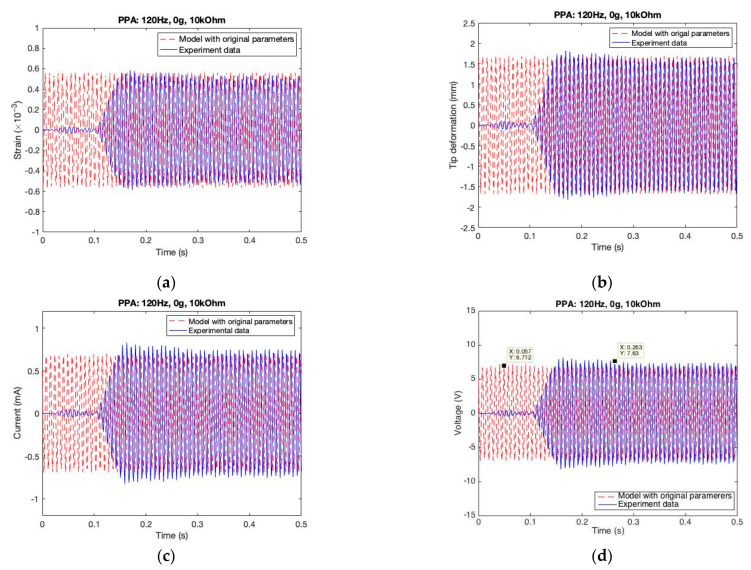
Comparison between the experimental and the model simulation results with original parameters in the PPA. (**a**) Strain vs. time; (**b**) Deformation vs. time; (**c**) Current vs. time; (**d**) Voltage vs. time; (**e**) Voltage vs. strain; (**f**) Power vs. strain.

**Figure 14 sensors-22-04457-f014:**
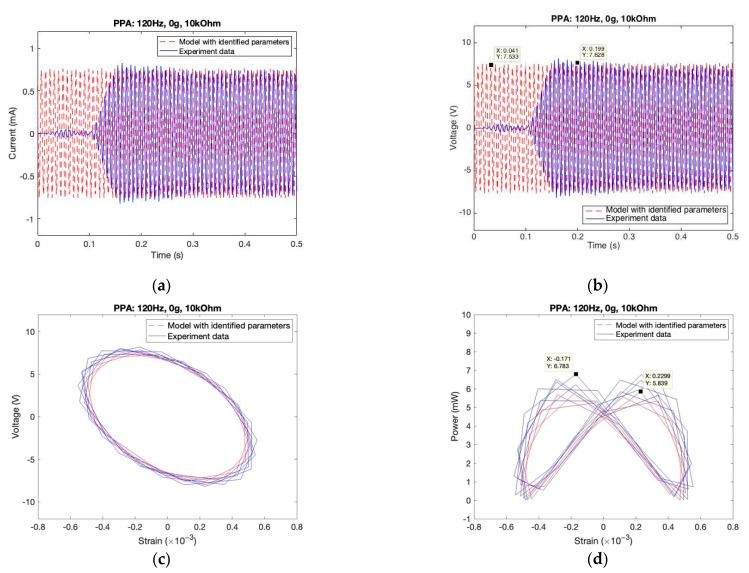
Comparison between experimental and the model simulation data with optimized parameters in PPA. (**a**) Current vs. time; (**b**) Voltage vs. time; (**c**) Voltage vs. strain; (**d**) Power vs. strain.

**Figure 15 sensors-22-04457-f015:**
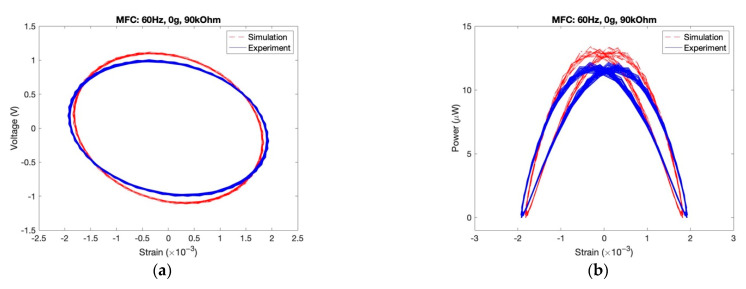
Comparison of experimental and simulation results in MFC. (**a**) Voltage vs. strain; (**b**) Power vs. strain.

**Figure 16 sensors-22-04457-f016:**
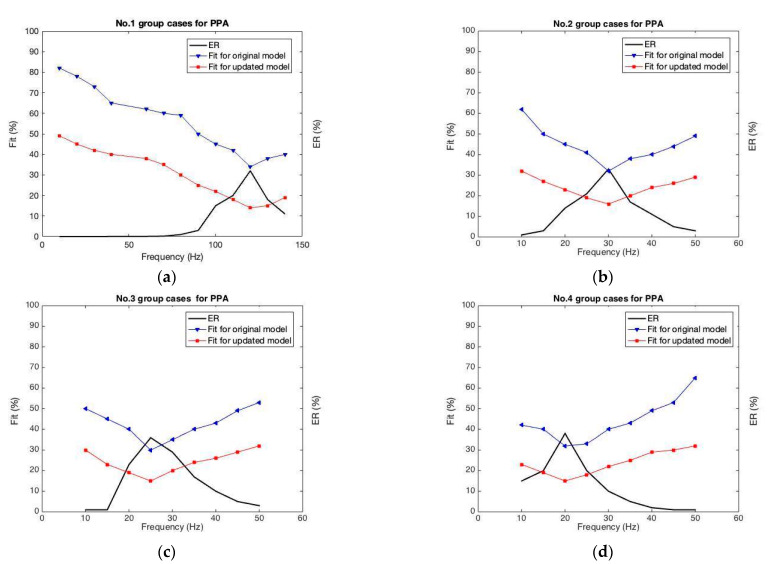
Error analysis of models with original and optimized parameters. (**a**) Error analysis for No. 1 group; (**b**) Error analysis for No. 2 group; (**c**) Error analysis for No. 3 group; (**d**) Error analysis for No. 4 group.

**Figure 17 sensors-22-04457-f017:**
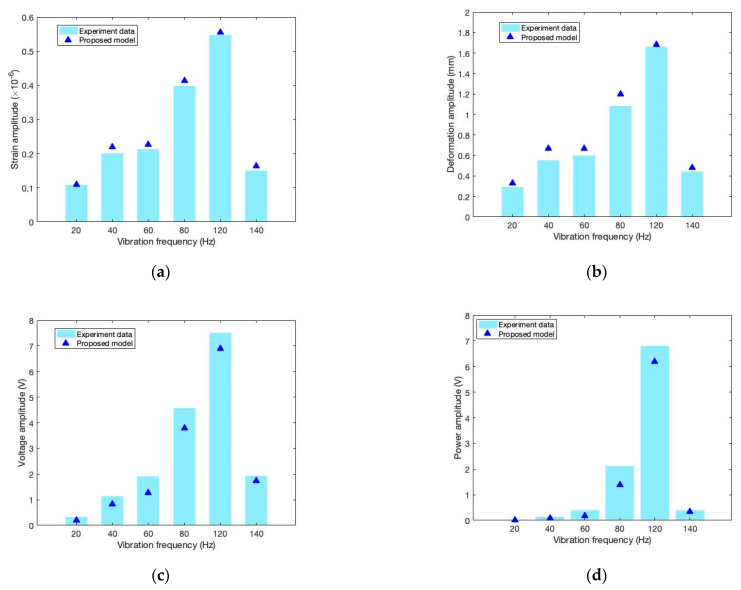
Changing law of responses’ amplitudes with vibration frequency. (**a**) Strain amplitude vs. vibration frequency; (**b**) Deformation amplitude vs. vibration frequency; (**c**) Voltage amplitude vs. vibration frequency; (**d**) Power amplitude vs. vibration frequency.

**Table 1 sensors-22-04457-t001:** Optimal settings of energy harvesters of MFC and PPA.

Structural Type	Harmonic Frequency	Optimal Resistance	Normalized Max/Mean Power
	(Hz)	(kOhm)	(mW)
MFC	64.85	-	6.9 μW /2.76 μW
PPA with 0 g mass	123.20	10	12.83/5.31
PPA with 12.5 g mass	31.20	30	7.16/2.94
PPA with 18.75 g mass	25.79	35	3.69/1.70
PPA with 25 g mass	21.94	50	2.32/0.98

**Table 2 sensors-22-04457-t002:** Original and optimized parameters of PPA and MFC for the electromechanical model.

Parameter	Unit	PPA		MFC	
		Original	Optimized	Original	Optimized
$h_{pc\;}$	μm	50	49.63	-	23.32
$h_{p}$	μm	150	129.74	-	0.5458
EI	$N\cdot {\mathrm{cm}}^{2}$	107	106.2	-	1.558
ϑ	mm	10	11.88	-	0. 8994

**Table 3 sensors-22-04457-t003:** Equivalent error analysis of the models with original and optimized parameters.

	Equivalent Error (EE) (%)				
	No. 1 Group	No. 2 Group	No. 3 Group	No. 4 Group	Average
Model with original parameters	38.77	42.12	46.24	41.37	42.04
Model with optimized parameters	16.92	21.83	25.19	21.2	21.85
EE improved by	56.36	48.17	45.52	48.76	48.03

## Data Availability

The data presented in this study are available on request from the corresponding author. The data are not publicly available due to privacy.

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
