# Peer review of "A Versatile Model for Describing Energy Harvesting Characteristics of Composite-Laminated Piezoelectric Cantilever Patches"

_sensors, 2022, doi:10.3390/s22124457_

Round 1

Reviewer 1 Report

This paper conducts a series of studies on energy harvesting by using two popular composite-laminated piezoelectric cantilever patches (MFC and PPA) in terms of experiment and modeling.

  1. Generally, theoretical modelling is firstly conducted and then the model is verified by the experiments. However, the experiments were firstly carried out in this paper. Then the model was validated with the experimental data in Section 5. The structure seemed not irrational. Why was such a structure of this paper arranged by the author? Whether could Section 2 be merged in Section 5?
  2. Followed by last comment, the author had better describe the section arrangement in the Introduction.
  3. As the author mentioned, the coupled electromechanical model was versatile and applicative for diversified harvesters with vague or even unknown parameters. Why was the model versatile? Besides, what kind of vague or even unknown parameters? Which unknown parameters were included, such as piezoelectric materials, piezoelectric constant, dimensions of piezoelectric elements, equivalent capacitance?
  4. Whether the coupled electromechanical model was only suitable for vibration-based energy harvester or more forms of energy?
  5. If the piezoelectric energy harvesters were not based on piezoelectric cantilever beam, whether the coupled electromechanical model was still correct? The description of this conclusion was not so rigors and clear.
  6. There are similar mistakes about the number of figures and “Error! Reference source not found” was shown. The errors should be corrected.

Author Response

Dear professor,

Thanks so much for reviewing our paper, here is comments (in black) and answer (in blue) as below. 

This paper conducts a series of studies on energy harvesting by using two popular composite-laminated piezoelectric cantilever patches (MFC and PPA) in terms of experiment and modeling.

1) Generally, theoretical modelling is firstly conducted and then the model is verified by the experiments. However, the experiments were firstly carried out in this paper. Then the model was validated with the experimental data in Section 5. The structure seemed not irrational. Why was such a structure of this paper arranged by the author? Whether could Section 2 be merged in Section 5?

Corrected: it is common seen that theoretical modeling should be written before the experiment. Here is the reason that we consider introducing experiment first as below.

This paper has studied one of most popular harvesters with composite-laminated piezoelectric cantilever patches. And it has been carried out by two main parts including experimental one and modeling one.

In section 2, two typical products (MFC and PPA) of this type harvesters are introduced, and their experiment are carried out. Based on that, we next conduct and propose one effective model to predict the harvested energy amount and imitate the characteristic behavior of the harvesters. It should be noted that the modeling method includes two processes. One is deducing a coupled electric-mechanical model with parameters in Section 3. And the other is to update the model with a set of optimized parameters for improving the model more versatile and practical in Section 4. Actually, the proposed model (section 3and 4) is presented on the basis of the harvesters and their experimental results such as the structural form, electric, mechanical parameters etc. For this reason, we first introduce the harvesters and their experiment (in section 2), next introduce modelling method next (in section 3 and 4).

2) Followed by last comment, the author had better describe the section arrangement in the Introduction.

Corrected: the introduction has been revised according to the comment, please see the revised manuscript.

3) As the author mentioned, the coupled electromechanical model was versatile and applicative for diversified harvesters with vague or even unknown parameters. Why was the model versatile? Besides, what kind of vague or even unknown parameters? Which unknown parameters were included, such as piezoelectric materials, piezoelectric constant, dimensions of piezoelectric elements, equivalent capacitance?

Corrected: In the case of laminated cantilever (see Figure 11), the coupled mechanical-electric model is proposed in section 3. There are several parameters in the model. Only if the parameters are assigned effectively, the model could be useful.

In fact, there exist a few of parameters in the model that is very hard to be assigned such as   (center-to-center interdigitated electrode spacing of the -th piezoelectric layer),  (equivalent distance from the neutral axis of the structural cross section to the center of piezoelectric layer),  (equivalent bending stiffness of the composite cross section for the constant electric field condition),  (coupling geometric coefficient). Those parameters are basically vague (comprehend-equivalent) and quite hard to assign them accurate values, which make the original model inaccurate even unusable. Those vague parameters are also explained in section 4.

In addition, many composite laminated harvesters are encapsulated whose parameters are too complicated to be extracted. Taking MFC for example, it has a complicated structure especially that the piezo layer isn’t single piezo material but fibrously embedded in polymer matrix (see Figure1(a)). It is quite challenging even impossible to extract the information of the indispensable parameters for the original model.

In view of aforementioned problems, we design a GA procedure to automatically optimize the effective values for those vague parameters (in section 4.2). by using the optimization procedure, the improved model can be effective used, no matter how vague of those parameters or how complicated structure of the harvest, which is versatile and applicative.

4) Whether the coupled electromechanical model was only suitable for vibration-based energy harvester or more forms of energy?

Corrected: The coupled electromechanical model is based on electric-mechanical theory. It is especially suitable for vibration-based energy harvester. If for other forms of energy, need to consider additional physical field like thermal, electromagnetic, etc.

5) If the piezoelectric energy harvesters were not based on piezoelectric cantilever beam, whether the coupled electromechanical model was still correct? The description of this conclusion was not so rigors and clear.

Corrected: This paper is mainly focused on investigating one of most popular harvesters that are laminated piezoelectric cantilever patches. And therefore, the proposed model is suitable for this kind of harvesters. The description of the conclusion was ambiguous. We have improved the ambiguity, please review the revised manuscript.

6) There are similar mistakes about the number of figures and “Error! Reference source not found” was shown. The errors should be corrected.

Corrected: According to the review comment, we checked and corrected all the errors.

Reviewer 2 Report

The paper is focused on the electromechanical investigation of a laminated piezoelectric cantilever and, specifically, it proposes and verifies a parametric model.

While the work is interesting, some issues need to be addressed. First of all, the review of the paper was very difficult because all the links to the figures are missing. Subsequently, the text reports many typos and a poor English language.

Here are my suggestions.

  • Section 1 needs to be improved. The cited literature is very limited, indeed the background of the study is not clearly described. As an example, some works, which focus on the electromechanical aspect, should be cited and discussed:

https://doi.org/10.1063/1.5093956

https://doi.org/10.1063/1.4952675

  • Line 46, check “Fabricated”;
  • Line 53, check “that uses”;
  • Lines 54-55, this aspect should be clarified;
  • Line 56, check the reference “Hagood et al [19]” which is not given in section 6;
  • Section 2.2 should be improved with more information. MFC and PPA patches are commercial devices and, for this reason, the productors should be indicated.
  • It is really important to report all the useful details so that the reader can repeat the experiments of the work. Then information (manufacturer, model ...) on the employed equipment (variable resistance, strain gauge ...) must be provided.
  • Lines 130-133, this sentence should rephrased;
  • Lines 145-147, this sentence should rephrased. Check the meaning of "magnetized", it does not sound good.
  • Lines 151-152, what was the frequency resolution of the sweep?
  • In Figures 4, 5, 6 etc, you need to improve the labels by describing all the parts (a, b, c…).
  • Line 271, check this label (Figure 1?);
  • In section 6 or in a specific "Discussion" section, I suggest to discuss better the improvements you have presented in this work, also comparing it with the state-of-the-art.

Author Response

Dear prof.,

Round 2

Reviewer 1 Report

Basically, the authors modified the  manuscript according to the reviewer's comments. Now it can be accepted.

Author Response

Dear prof.,

Thanks so much for your precious time. Best wishes

Xiaomin Xue

Reviewer 2 Report

I recommend that the Authors focus their attention on the introduction and on the discussion of the experimental results.

Specifically, the Authors report 11 references on the basic knowledge of energy harvesting, but none concerns the deformation behavior of piezoelectric energy harvesters.

For this aim, I suggested two articles (https://doi.org/10.1063/1.5093956 and https://doi.org/10.1063/1.4952675), which are also useful for discussing their experimental results, but they have not been taken into account.

Author Response

Dear prof.,

Thanks so much for your review comment. In light of your suggestions, we have revised it and submit the updated one, and the "notes" for you is at the attachment. Please check it, thank you.

Best wishes

Xiaomin Xue
